# CLIP-MoE: Towards Building Mixture of Experts for CLIP with Diversified Multiplet Upcycling

## Abstract

In recent years, Contrastive Language-Image Pre-training (CLIP) has become a cornerstone in multimodal intelligence. However, recent studies have identified that the information loss in the encoding process of CLIP is substantial. Such deficiency significantly limits the ability of a single CLIP model to handle images rich in visual detail. In this work, we propose a simple yet effective model-agnostic strategy, **Diversified Multiplet Upcycling (DMU)** for CLIP. It integrates multiple CLIP models that capture diversified, complementary information into a Mixture of Experts (MoE) architecture. Inspired by the recently proposed Multistage Contrastive Learning (MCL), which constructs multiple CLIP models that share the same structure while capturing different complementary information, Diversified Multiplet Upcycling efficiently fine-tunes a series of CLIP models from a dense pre-trained CLIP checkpoint to capture different feature distributions, sharing parameters except for the Feed-Forward Network (FFN). These models are then transformed into a **CLIP-MoE** with a larger model capacity but minimal computational overhead. Extensive experiments demonstrate the significant performance of CLIP-MoE across various zero-shot retrieval, zero-shot image classification tasks, and downstream Multimodal Large Language Model (MLLM) benchmarks by serving as a vision encoder. Furthermore, Diversified Multiplet Upcycling enables the conversion of any dense CLIP model into CLIP-MoEs, which can seamlessly replace CLIP in a plug-and-play manner without requiring further adaptation in downstream frameworks. Through Diversified Multiplet Upcycling, we aim to provide valuable insights for future research on developing more efficient and effective multimodal learning systems.

## 1 Introduction

Contrastive Language-Image Pre-training (CLIP) (Radford et al., 2021) is a strong vision-language foundation model that utilizes large-scale datasets to learn comprehensive visual representations by bridging vision and language via contrastive image-text pre-training. Beyond traditional tasks like classification, CLIP has been broadly applied in areas such as image (Wang et al., 2023; Zhang et al., 2023), audio (Guzhov et al., 2022), and video (Rasheed et al., 2023) understanding, cross-modal retrieval (Ma et al., 2022; Zhao et al., 2024), multimodal generation (Ramesh et al., 2022; Xie et al., 2024), and data filtering (Schuhmann et al., 2022). Additionally, CLIP serves as the vision encoder for various Multimodal Large Language Models (MLLMs) (Alayrac et al., 2022; Liu et al., 2024b;c; Chen et al., 2024b; Li et al., 2024c).

However, existing CLIP models still face inherent limitations. Recent studies have highlighted that CLIP often encodes inputs in a very coarse-grained manner, overlooking much useful information (Tang et al., 2023; Tong et al., 2024b; Bleeker et al., 2022). As a result, CLIP frequently produces blind pairs (Tong et al., 2024b), where two semantically different images with similar visual components are encoded into the same representation. This leaves downstream models with insufficient information, especially when CLIP serves as a vision encoder. Such substantial information loss negatively impacts downstream tasks and can confuse downstream models, such as the base LLMs in Multimodal Large Language Models (MLLMs). To address this issue and enhance CLIP's ability to encode richer information, efforts have been made to improve the quality of training data

and to scale up model size. However, these works typically involve retraining the CLIP model from scratch (Li et al., 2024b; Ma et al., 2024; Xu et al., 2023), which is both resource-intensive and costly. Additionally, there are attempts to ensemble different types of vision encoders (Tong et al., 2024b; Shi et al., 2024), which makes the entire model heterogeneous and total parameters grow explosively.

To address the above limitations, we propose a simple yet effective model-agnostic strategy, **Diversified Multiplet Upcycling (DMU)**, for CLIP, which leverages the sparsely activated Mixture of Experts (MoE) framework to extend model capacity while fully utilizing off-the-shelf pre-trained dense checkpoints, avoiding the need for training from scratch. MoE has proven effective in scaling large pre-trained models by using fixed activated parameters, enhancing both performance and robustness (Jiang et al., 2024; Dai et al., 2024; Chen et al., 2024a). In Diversified Multiplet Upcycling, we first fine-tune the base dense CLIP model to produce a series of multiplet CLIP models using the recently proposed Multistage Contrastive Learning (MCL) (Zhang et al., 2024b). MCL generates models that encode diversified information through a multistage clustering and fine-tuning process. By **multiplet**, we refer to CLIP models that share all parameters except for the feed-forward network (FFN) layers during MCL fine-tuning. By **diversified**, we mean that these models yield a series of FFN experts, each capturing different aspects of the input information, which are then used to initialize a **CLIP-MoE** model. Finally, through fine-tuning the router in CLIP-MoE, we ensure the full utilization of all experts, enabling CLIP-MoE to capture richer and more useful information than the base model, while leveraging sparse activation to avoid the explosion of activated parameters.

We demonstrate that using a small high-quality image-caption dataset, our MCL-initialized CLIP-MoE significantly improves CLIP's performance. Notably, on retrieval tasks, CLIP-MoE outperforms the base OpenAI CLIP model by about 20%, while incurring minimal additional training overhead—less than 2% of the total computational cost of training the base CLIP model from scratch. When serving as a vision encoder for MLLMs, CLIP-MoE also shows substantial improvements in most benchmarks simply by replacing the original vision encoder. Our experiments show that CLIP-MoE not only outperforms other fine-tuning baselines but also surpasses popular MoE-construction methods like Sparse Upcycling (Komatsuzaki et al., 2022). To the best of our knowledge, this work is the first to introduce sparsely activated MoE into CLIP foundation models, whereas previous methods have focused either on vision representation (Li et al., 2024a) or model-wise ensembling (Ma et al., 2024).

In summary, the contributions of this work are as follows: *First*, we propose a novel method, Diversified Multiplet Upcycling for CLIP, which initializes a CLIP-MoE using FFN experts obtained through multistage fine-tuning, offering a new pathway to effectively scale the CLIP foundation model. *Second*, we demonstrate that our model-agnostic Diversified Multiplet Upcycling significantly improves model performance by fully leveraging new high-quality data and pre-trained CLIP checkpoints, while avoiding the high computational costs associated with training from scratch. *Third*, we conduct extensive experiments, showing that our upcycled CLIP-MoE achieves significant performance improvements over the original CLIP and other baselines with lower computational costs across various downstream tasks, including classification, retrieval, and serving as a vision encoder for MLLMs.

## 2 RELATED WORKS

### 2.1 CONTRASTIVE LEARNING

In contrastive learning, the core objective is to minimize the distance between positives and the anchor while maximizing the distance between negatives and the anchor within the representation space. This objective compels the model to effectively encode sufficient information of the inputs to distinguish anchors from their negatives.

Contrastive learning has become a central technique in self-supervised learning, aiming to learn representations by bringing semantically similar samples closer in the embedding space while pushing dissimilar samples apart (Chen et al., 2020; He et al., 2020). This approach has been particularly successful in multimodal settings, where models like Contrastive Language-Image Pre-training (CLIP) (Radford et al., 2021) have emerged as foundational tools. CLIP aligns visual and textual

representations by training on vast datasets of paired images and text, enabling the model to bridge different modalities effectively.

Despite its success, CLIP is not without its limitations. One significant shortcoming is its tendency to encode only coarse-grained visual concepts, which can lead to the loss of fine-grained information that is crucial for certain downstream tasks (Tang et al., 2023; Tong et al., 2024b). To address these limitations, recent works mainly focus on improving the quality of training data (Li et al., 2024b; Ma et al., 2024; Xu et al., 2023; Zhang et al., 2024a). However, most of these approaches require retraining the model from scratch, which is computationally expensive, time-consuming, and not easily extendable when better data becomes available.

## 2.2 MIXTURE-OF-EXPERTS

The Mixture-of-Experts (MoE) architecture could scale the model capacity without additional computational cost (Fedus et al., 2022a). For each input token, only top-$k$ best experts are selected to obtain an aggregated representation (Shazeer et al., 2017). This sparsity allows MoE models to scale to trillions of parameters while maintaining the computational efficiency (Lepikhin et al., 2020; Fedus et al., 2022b). Due to the large model capacity, the performance could be improved by large margins (Rajbhandari et al., 2022; Dai et al., 2024). Besides, specialized experts in MoE models are good at handling a wide range of tasks (Shen et al., 2023; Zhu et al., 2024; Lu et al., 2024) with high robustness (Chen et al., 2024a).

However, one challenge in MoE training is expert initialization. Sparse Upcycling (Komatsuzaki et al., 2022) has been proposed as a technique to initialize MoE models by copying Feed-Forward Networks (FFN) from dense models as multiple experts. It selectively activates and fine-tunes only a sparse subset of parameters. This method significantly reduces the training cost.

In this work, we explore the integration of Multistage Contrastive Learning (MCL) with the MoE architecture. By using MCL to initialize the experts, we aim to capture complementary information across different CLIP experts, which can then be leveraged by the MoE structure to enhance overall performance with minimal additional computational cost.

## 3 PRELIMINARIES

### 3.1 MULTISTAGE CONTRASTIVE LEARNING (MCL)

Multistage Contrastive Learning (MCL) (Zhang et al., 2024b) is designed to obtain a series of contrastive models, each capturing different and complementary information from the input data through multiple cluster-and-contrastive processes. Specifically, at each stage, the learned representations are clustered. In the following stage, for each anchor, negative samples are drawn only from the same accumulated cluster from the previous stages. In this way, the model learns new information beyond what was captured in earlier stages. For example, consider a dataset containing objects with varying shapes, colors, and textures. In the first stage, the contrastive model might focus on learning color information. After clustering, samples within the same cluster will share the same color. In the second stage, since the anchor and its negative samples share the same color, the model is compelled to learn other features, such as texture, to differentiate between them. After clustering in the second stage, samples in the same accumulated cluster will now share both color and texture. Consequently, in the third stage, the model must focus on other attributes, such as shape, to distinguish between samples. After three stages, we obtain three contrastive models, each encoding distinct information: color, texture, and shape.

Formally, let $\boldsymbol{X} = \{\mathbf{x}_i\}_{i=1}^{M}$ represent a dataset. After training the encoder in the first stage, we obtain encoded representations $\boldsymbol{Z}_0 = \{f_0(\mathbf{x}_i)\}_{i=1}^{M}$. By clustering $\boldsymbol{Z}_0$, we obtain cluster assignments $\boldsymbol{Y}_0 = \{\mathbf{y}_{(i,0)}\}_{i=1}^{M}$. In the $j^{th}$ stage, after the cluster-and-contrastive process, each sample $\mathbf{x}_i$ is assigned to an accumulated cluster $\hat{\mathbf{y}}_{(i,j)} = [\mathbf{y}_{(i,0)}, \cdots, \mathbf{y}_{(i,j-1)}]$. The objective at the $j^{th}$ stage is:

$$\mathcal{L} = \mathbb{E}_{\mathbf{x},\mathbf{x}^+,\{\mathbf{x}_i^- | \hat{\mathbf{y}}_j = \hat{\mathbf{y}}_{(i,j)}^-\}_{i=1}^{m}} \left[ -\log \frac{e^{s(\mathbf{z},\mathbf{z}^+)/\tau}}{e^{s(\mathbf{z},\mathbf{z}^+)/\tau} + \sum_{i=1}^{m} e^{s(\mathbf{z},\mathbf{z}_i^-)/\tau}} \right], \tag{1}$$

where $\hat{\mathbf{y}}_j$ represents the accumulated cluster assignment of the anchor $\mathbf{x}$ at the $j^{th}$ stage; $\hat{\mathbf{y}}^-_{(i,j)}$ denotes the accumulated cluster assignment of the negative sample $\mathbf{x}^-_i$ at the $j^{th}$ stage; and $s(\cdot, \cdot)$ denotes cosine similarity. In our proposed Diversified Multiplet Upcycling, we leverage the MCL framework to fine-tune a base model and extract a series of experts for the MoE, whereas the original MCL results in a series of standalone CLIP models.

## 3.2 MIXTURE OF EXPERTS (MoE)

Mixture of Experts (MoE) is an efficient architecture designed to scale large models by dynamically routing inputs through a subset of specialized sub-models, or "experts". This structure allows the model to maintain high overall capacity while only utilizing a fraction of its parameters for any given input, thereby optimizing both computational efficiency and performance.

In the context of Transformer, an MoE layer (Jiang et al., 2024) typically replaces the standard feed-forward network (FFN) with a set $\{E_i\}_{i=1}^N$ of $N$ experts, each of which is an independent FFN. Given an input token representation $\mathbf{x}$, it first passes through a gating network $\mathbf{W}_r$ to obtain the logits corresponding to each expert, then the largest Top-K experts will be chosen, and finally, the probabilities of these selected experts are normalized using Softmax. In this way, we can obtain the probability $R(\mathbf{x})$ of selected experts among all $N$ experts. Notably, the probability of non-

$$\mathbf{x}_{\text{out}} = \sum_{i=1}^N R(\mathbf{x})_i \cdot E_i(\mathbf{x}), \quad R(\mathbf{x}) = \text{Softmax}(\text{TopK}(\mathbf{x} \cdot \mathbf{W}_r)), \tag{2}$$

where $R(\mathbf{x})_i$ denotes the $i$-th routing weight vector produced by the router network $\mathbf{W}_r$.

To ensure that all experts are utilized effectively and prevent the model from overfitting to a small subset of experts, a load balancing loss (Fedus et al., 2022b) is often added to the primary loss function. This loss penalizes imbalanced expert usage by encouraging a more uniform distribution of the input tokens across all experts.

## 4 DIVERSIFIED MULTIPLET UPCYCLING FOR CLIP

### 4.1 EXPERT EXTRACTION

We begin by extracting a series of Feed-Forward Network (FFN) layers utilizing Multistage Contrastive Learning (MCL) to fine-tune a pre-trained base CLIP model for multiple stages. During fine-tuning, we freeze all parameters of the base CLIP model except for the FFN layers within each transformer block in both the image and text encoders. Because the distributions of contrastive negative samples in different MCL stages are distinct, the FFN layers at each stage will learn diversified and complementary information distinct from previous stages. For clarity, we use superscripts to index the transformer blocks and subscripts to index the MCL stages or MoE experts. Suppose we are fine-tuning a transformer-based CLIP model, where the image encoder contains $A$ transformer blocks and the text encoder contains $B$ transformer blocks. The FFN layers in the original base model are denoted as $\{E_0^{(i)}\}_{i=1}^{A+B}$. As illustrated in Figure 1, the base model might initially focus on color-related information. During MCL Stage 1, only the FFN layers are fine-tuned. After the cluster-and-contrast process in MCL, the FFN layers $\{E_1^{(i)}\}_{i=1}^{A+B}$ in the fine-tuned model learn new information beyond color, such as texture. In MCL Stage 2, the model further fine-tunes the FFN layers, resulting in $\{E_2^{(i)}\}_{i=1}^{A+B}$, which now encodes additional features such as shape. Through two stages of MCL, we obtain FFN layers where $\{E_0^{(i)}\}_{i=1}^{A+B}$ focus on color, $\{E_1^{(i)}\}_{i=1}^{A+B}$ on texture, and $\{E_2^{(i)}\}_{i=1}^{A+B}$ on shape.

### 4.2 INITIALIZATION OF MIXTURE OF EXPERTS

Once a series of FFN layers $\{E_j^{(i)}\}_{j=0}^N$ have been obtained through $N$ stages of MCL, we utilize these FFNs as the experts in a Mixture of Experts (MoE) model, as depicted in Figure 1. According

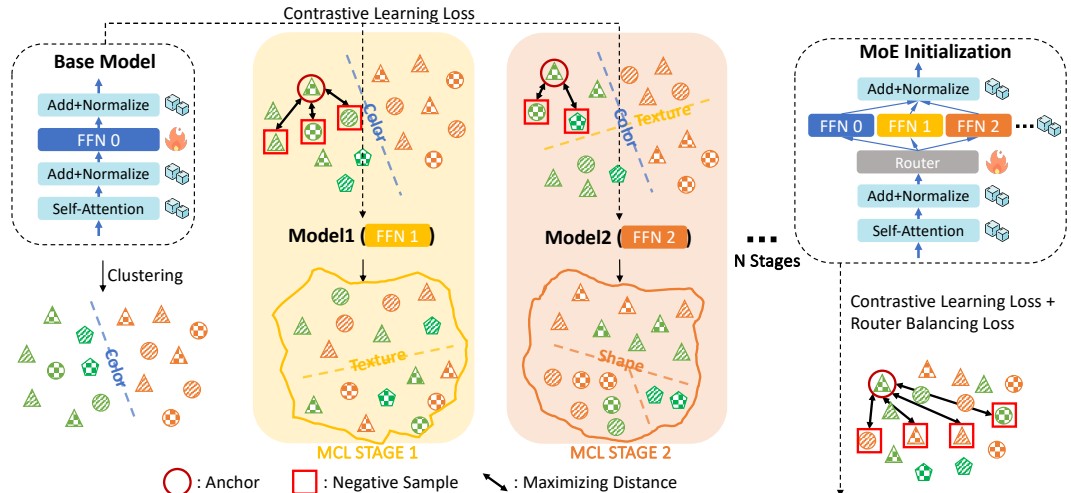

Figure 1: Overview of Diversified Multiplet Upcycling: Our approach involves three key steps. (a) Fine-tuning the base CLIP model using the MCL framework while freezing all parameters except for the FFN layers. This process yields a new set of FFN layers at each stage of MCL. (b) Using the obtained FFN layers as experts to initialize a CLIP-MoE. (c) Continuously fine-tuning the CLIP-MoE using both contrastive learning loss and a router balancing loss to optimize the routers. The terms 'color', 'shape', and 'texture' are metaphorical representations of abstract features.

to Equation 2, in the $i^{th}$ transformer block of the base CLIP model, the original FFN layer is replaced with a randomly initialized router and a set of experts:

$$\mathbf{x}_{\text{out}}^{(i)} = \sum_{j=0}^{N} R^{(i)}(\mathbf{x}^{(i)})_j \cdot E_j^{(i)}(\mathbf{x}^{(i)}), \quad R^{(i)}(\mathbf{x}^{(i)}) = \text{Softmax}(\text{TopK}(\mathbf{x}^{(i)} \cdot \mathbf{W}_r^{(i)})), \qquad (3)$$

where $R^{(i)}(\mathbf{x})_j$ denotes the $j$-th component of the routing weight vector produced by the router network $\mathbf{W}_r^{(i)}$ in the $i^{th}$ transformer block. This setup results in a CLIP-MoE model where different experts within different transformer blocks specialize in distinct aspects of the input.

## 4.3 CONTINUOUS FINE-TUNING OF CLIP-MoE

To enable the model to learn optimal routing strategies while preserving the information learned by the FFN layers during MCL, we further fine-tune the routers while freezing all other parameters. We apply the standard contrastive learning loss while incorporating an auxiliary load balancing loss, following the approach from Fedus et al. (2022b), to encourage a balanced load across experts. Given $N + 1$ experts indexed by $j = 0$ to $N$, and a batch $\mathcal{B}$ with $T$ tokens, the load balancing loss for the $i^{th}$ transformer block is defined as:

$$\mathcal{L}_{balance} = N \cdot \sum_{j=0}^{N} f_j \cdot P_j, \ \ f_j = \frac{1}{T} \sum_{x \in \mathcal{B}} \mathbb{1}\{\arg\max p(x) = j\}, \ \ P_j = \frac{1}{T} \sum_{x \in \mathcal{B}} p_j(x), \qquad (4)$$

where $f_j$ is the fraction of tokens assigned to expert $j$, and $p(x)$ is the logits output from the router network; $P_j$ represents the fraction of router probability allocated to expert $j$, which is the mean of $p_j(x)$, the probability of routing token $x$ to expert $j$. For simplicity, we omit the transformer block index $i$ in the equation. Since $f_j$ and $P_j$ are positive and both their sums are equal to 1, $\mathcal{L}_{balancing}$ is minimized if and only if $f_j = \frac{1}{T}$, $P_i = \frac{1}{T}$. This balancing loss encourages not only a uniform distribution of actual tokens routed to each expert (i.e., ensuring that all experts have equal importance), but also a uniform distribution of router confidence across tokens (i.e., preventing the router from being overly confident for some tokens and underconfident for others). With this

auxiliary load balancing loss, the total loss is given by:

$$\mathcal{L} = \mathcal{L}_{CLIP} + \alpha \cdot \frac{1}{A+B} \sum_{i=1}^{A+B} \mathcal{L}_{balance}^{(i)}.$$ (5)

Following Fedus et al. (2022b), we set $\alpha = 0.01$ by default. By applying MoE-Packing to CLIP, we obtain a CLIP-MoE model that is capable of capturing more useful information than the base model, with minimal computational overhead, resulting in a robust and efficient enhancement of CLIP.

## 5 EXPERIMENTS

### 5.1 DATASETS

To fully showcase the potential of our MCL-initialized CLIP-MoE, we implement our experiments on the following two image-caption datasets respectively.

**Recap-DataComp.** Recap-DataComp-1B (Li et al., 2024b) is a large-scale dataset comprising 1.3 billion high-quality image-caption pairs. This dataset is derived from the original DataComp-1B dataset, with all images re-captioned using a fine-tuned LLaVA-1.5 model powered by LLaMA-3 (Dubey et al., 2024). Li et al. (2024b) utilized this dataset to train CLIP models from scratch, resulting in significant improvements in retrieval performance. Due to computational constraints, our experiments use a randomly sampled subset of 1 million pairs from Recap-DataComp-1B, referred to as Recap-DataComp-1M, to demonstrate the data efficiency of our proposed pipeline.

**ShareGPT4V.** ShareGPT4V (Chen et al., 2023) is a high-quality image-text dataset containing 1.2 million highly descriptive captions. The captions are generated by a Multimodal Large Language Model (MLLM) fine-tuned on 100k image-text pairs produced by GPT4V, resulting in well-aligned image-text pairs.

### 5.2 BASELINES

**Direct Fine-tuning.** As our experiments incorporate additional data, we use direct fine-tuning as a basic baseline to evaluate the performance contributions from the additional data.

**Sparse Upcycling.** Sparse Upcycling (Komatsuzaki et al., 2022) is a widely adopted method for initializing a Mixture of Experts (MoE) model using a pre-trained dense checkpoint. It is a simple yet effective approach for scaling up a pre-trained model and is much more efficient than training an MoE from scratch.

**Long-CLIP.** Long-CLIP (Zhang et al., 2024a) introduces an efficient pipeline to enhance CLIP performance through fine-tuning on high-quality image-caption datasets with long captions. It aligns the long caption of an image with the encoded image features and the short caption with the primary components of the image features. While effective on the ShareGPT4V dataset, Long-CLIP is limited to datasets with a similar structure, where each image has both one short and one long caption. Moreover, it requires significantly more computational resources compared to our approach.

**LLaVA-1.5.** LLaVA-1.5 (Liu et al., 2024a) is an improved version of LLaVA (Liu et al., 2024b), commonly used as a baseline for MLLMs. It bridges a pre-trained CLIP vision encoder with a pre-trained LLM using a simple MLP, enabling the LLM to gain visual understanding with minimal fine-tuning on image-text pairs. We evaluate the representation quality of our CLIP-MoE by replacing the vision encoder in the original LLaVA-1.5 with our CLIP-MoE and fine-tuning it following the same pipeline as LLaVA-1.5.

### 5.3 TRAINING SETUP

By default, we use OpenAI CLIP-ViT-L/14 (Radford et al., 2021) as the base model for our Diversified Multiplet Upcycling approach. During the clustering process at each stage of MCL, we cluster the image features into 3 clusters and the text features into 3 clusters, resulting in 9 clusters per stage (the Cartesian product of the image and text feature clusters). To accommodate longer text inputs, we interpolate the positional embeddings following the approach in (Zhang et al., 2024a).

The global batch size is maintained at 800 unless otherwise specified. To balance performance and computational cost, we set the number of experts to 4 and use top-2 activation.

## 5.4 TRAINING COST

We use 8 A100 GPUs for training. To train the CLIP-MoE model with four experts, we introduce three additional MCL fine-tuning stages, each trained for 1 epoch. When using the ShareGPT4V dataset, each MCL stage takes approximately 0.5 hours, and the router fine-tuning stage also takes about 0.5 hours. In total, the training time is less than 2.5 hours. In comparison, Long-CLIP training under the same conditions takes around 6 hours, making our approach significantly more efficient. Our maximum GPU memory usage is 8×65955MB, which is comparable to Long-CLIP's 8×63581MB. When training on the Recap-DataComp-1M dataset, the training cost is even lower. During inference, with top-2 activation, the activated parameter size of our CLIP-MoE is approximately 1.7 times that of the base model (OpenAI CLIP-ViT-L/14).

## 5.5 EVALUATION

We begin by evaluating the performance of CLIP-MoE on Zero-Shot Image-Text Retrieval, a key task for assessing whether the CLIP model can capture rich fine-grained information, following Zhang et al. (2024a). All baselines are trained and compared using the Recap-DataComp-1M (Recap-DC) and ShareGPT4V (ShareGPT) datasets, with the exception of Long-CLIP. Long-CLIP is incompatible with the Recap-DataComp dataset, as it requires both a short and long caption for each image, whereas Recap-DataComp provides only one caption per image. Next, we assess the effectiveness of CLIP-MoE as a vision encoder within LLaVA-1.5, a representative Multimodal Large Language Model (MLLM). LLaVA-1.5 serves as an effective visual representation evaluator, helping to mitigate potential biases present in traditional evaluation tasks (Tong et al., 2024a). Finally, we test CLIP-MoE on traditional Zero-Shot Image Classification tasks, which rely more on coarse-grained features.

**Zero-Shot Image-Text Retrieval.** Following the methodology outlined in Zhang et al. (2024a), we

| Dataset | Model | COCO I2T | | | COCO T2I | | | Flickr I2T | | | Flickr T2I | | |
|---------|-------|------|------|------|------|------|------|------|------|------|------|------|------|
| | | @1 | @5 | @10 | @1 | @5 | @10 | @1 | @5 | @10 | @1 | @5 | @10 |
| | OpenAI | 56.1 | 79.5 | 86.8 | 35.4 | 60.1 | 70.2 | 48.5 | 72.6 | 80.8 | 28.0 | 49.3 | 58.7 |
| **Recap-DC** | Direct FT | 58.9 | 81.5 | 88.5 | 44.3 | 69.5 | 78.8 | 41.6 | 66.5 | 76.1 | 37.2 | 60.4 | 69.5 |
| | Upcycling | 59.2 | 81.7 | 88.7 | 45.8 | 70.9 | 79.9 | 42.1 | 67.3 | 77.0 | 39.4 | 62.9 | 71.7 |
| | CLIP-MoE | **64.0** | **85.1** | **90.8** | **45.2** | **70.2** | **79.4** | **56.8** | **80.1** | **87.0** | **40.8** | **63.8** | **72.5** |
| **ShareGPT** | Direct FT | 63.3 | 84.9 | 91.0 | 44.5 | 70.0 | 78.9 | 50.5 | 74.4 | 82.3 | 38.5 | 61.3 | 69.9 |
| | Upcycling | 62.9 | 84.6 | 90.8 | 45.2 | 70.6 | 79.6 | 49.6 | 73.8 | 82.1 | 39.5 | 62.4 | 71.1 |
| | Long-CLIP | 62.8 | 85.1 | 91.2 | 46.3 | 70.8 | 79.8 | 53.4 | 77.5 | 85.3 | 41.2 | 64.1 | 72.6 |
| | CLIP-MoE | **65.0** | **86.0** | **92.0** | **46.8** | **71.7** | **80.4** | **60.5** | **82.3** | **88.8** | **42.1** | **64.7** | **73.2** |

Table 1: Performance comparison on image-to-text (I2T) and text-to-image (T2I) retrieval tasks using the COCO and Flickr30k datasets. The models were trained and evaluated on the Recap-DataComp-1M (Recap-DC) and ShareGPT4V (ShareGPT) datasets, respectively. The best performance for each dataset is highlighted in bold. Our proposed CLIP-MoE consistently outperforms all baselines across all tasks.

evaluate text-to-image (T2I) and image-to-text (I2T) retrieval on the 5k COCO validation set (Lin et al., 2014) and the 30k Flickr30k (Young et al., 2014) dataset. The results are presented in Table 1. Given that both Recap-DataComp-1M and ShareGPT4V datasets offer higher caption quality and longer average caption lengths compared to web datasets, Direct Fine-Tuning, Sparse Upcycling, and CLIP-MoE demonstrate superior performance over the original OpenAI model across most tasks, including COCO I2T, COCO T2I, and Flickr T2I. However, for Flickr I2T, Sparse Upcycling, and Direct Fine-Tuning show significant performance degradation on the Recap-DC dataset. In this fine-tuning context, Sparse Upcycling only provides a limited advantage over Direct Fine-Tuning. Although Long-CLIP clearly outperforms both Direct Fine-Tuning and Sparse Upcycling,

it is incompatible with the Recap-DataComp dataset, because it requires each image to have both a short and a long caption. In contrast, our proposed CLIP-MoE surpasses all baselines on both Recap-DataComp and ShareGPT4V, maintaining consistent performance by leveraging the diverse information extracted by MoE experts initialized through different stages of MCL.

**Performance in LLaVA-1.5**

We further evaluate CLIP-MoE as the vision encoder within the LLaVA-1.5 model. The original vision encoder for LLaVA-1.5 is OpenAI's CLIP-ViT-L/14@336px (Radford et al., 2021), which is trained on images with a resolution of 336x336 pixels. To ensure a fair comparison, we use OpenAI's CLIP-ViT-L/14@336px as the base model for MCL and train our CLIP-MoE on the ShareGPT4V dataset at the same 336x336 resolution. After obtaining CLIP-MoE, we freeze it as the vision encoder and follow the same two-stage training procedure as LLaVA-1.5, using Vicuna-7B (Chiang et al., 2023) as the base LLM for CLIP-MoE-LLaVA1.5-7B and Vicuna-13B (Chiang et al., 2023) as the base LLM for CLIP-MoE-LLaVA1.5-13B. The evaluation results, shown in Table 2, demonstrate that by simply replacing the vision encoder with our CLIP-MoE, the final MLLM achieves significant performance improvements across most downstream tasks. This supports the conclusion that our CLIP-MoE is capable of extracting more useful information from image inputs and encoding higher-quality image representations.

| Method | MME | VQAv2 | TextVQA | POPE | MMBench |
|---|---|---|---|---|---|
| LLaVA1.5-7B | **1510.7** | 78.5 | 58.2 | 85.9 | 64.3 |
| CLIP-MoE-LLaVA1.5-7B | 1486.2 | **79.2** | **58.8** | **86.4** | **66.1** |
| LLaVA1.5-13B | 1531.3 | **80.0** | **61.3** | 85.9 | 67.7 |
| CLIP-MoE-LLaVA1.5-13B | **1593.7** | **80.0** | 60.9 | **86.3** | **69.1** |

Table 2: Performance comparison between OpenAI CLIP and CLIP-MoE as vision encoders in LLaVA1.5. The best performance for each dataset is highlighted in bold.

**Zero-Shot Image Classification.** We evaluated the zero-shot image classification accuracy on Ima-

| Dataset | Model | ImageNet | ImageNet-O | ImageNet-V2 | Cifar10 | Cifar100 | Avg. |
|---|---|---|---|---|---|---|---|
| | OpenAI | **75.5** | 31.9 | **69.9** | 95.4 | 76.8 | 69.9 |
| **Recap-DC** | Direct FT | 57.0 | **32.8** | 51.3 | 91.6 | 68.7 | 60.3 |
| | Upcycling | 61.1 | 32.3 | 55.3 | 93.6 | 71.0 | 62.7 |
| | CLIP-MoE | 74.3 | 32.2 | 68.7 | **95.5** | **79.3** | **70.0** |
| **ShareGPT** | Direct FT | 59.8 | **34.5** | 53.3 | 87.8 | 63.1 | 59.7 |
| | Upcycling | 62.5 | 34.4 | 56.5 | 91.3 | 67.5 | 62.5 |
| | Long-CLIP | 73.5 | 33.7 | 67.9 | 95.3 | 78.5 | 69.8 |
| | CLIP-MoE | 74.6 | 33.5 | 68.5 | **95.7** | **79.6** | **70.4** |

Table 3: Performance comparison on zero-shot image classification. The models were trained and evaluated on the Recap-DataComp-1M (Recap-DC) and ShareGPT4V (ShareGPT) datasets, respectively. The best performance for each dataset is highlighted in bold. CLIP-MoE achieved the highest average performance across both Recap-DC and ShareGPT.

geNet (Deng et al., 2009), ImageNet-O (Hendrycks et al., 2021), ImageNet-V2 (Recht et al., 2019), CIFAR-10 (Krizhevsky et al., 2009), and CIFAR-100 (Krizhevsky et al., 2009). The results are shown in Table 3. Both Direct Fine-Tuning and Sparse Upcycling exhibited significant performance degradation across most classification tasks, which is consistent with the observations in Zhang et al. (2024a). This decline in performance may be attributed to model overfitting, as both the Recap-DataComp and ShareGPT4V datasets contain approximately 1 million samples, a substantially smaller dataset compared to the 400M samples used for training OpenAI's CLIP. While Direct Fine-Tuning and Sparse Upcycling successfully learned more fine-grained information from the improved and lengthier image captions, leading to enhanced retrieval performance, they also lost the original model's ability to encode more coarse-grained information, resulting in decreased classification accuracy. In contrast, our proposed CLIP-MoE demonstrated a superior ability to preserve

classification performance compared to Long-CLIP and even surpassed the original OpenAI CLIP on ImageNet-O, CIFAR-10, and CIFAR-100. Additionally, CLIP-MoE achieved the best average performance when trained on both Recap-DC and ShareGPT datasets.

## 5.6 DISCUSSION

**Ablation Study on MCL**

To further validate the effectiveness of expert extraction utilizing MCL in Diversified Multiplet Upcycling, we conducted an ablation study on ShareGPT4V by training a CLIP-MoE model with only two experts: one from the original OpenAI CLIP and one from fine-tuning FFN layers on ShareGPT4V. As seen in Table 4, the performance of CLIP-MoE on the retrieval tasks is consistently higher than the model without MCL stages 1 and 2, demonstrating that more MCL stages do obtain experts that capture more useful information. The slight degradation in ImageNet zero-shot classification performance is expected, as not all of the additional learned information is beneficial for classification, which tends to rely on more coarse-grained features (Zhang et al., 2024a).

| Method | ImageNet Top-1 | COCO I2T @1 | @5 | @10 | COCO T2I @1 | @5 | @10 | Flickr I2T @1 | @5 | @10 | Flickr T2I @1 | @5 | @10 |
|--------|---------|-----|-----|-----|-----|-----|-----|-----|-----|-----|-----|-----|-----|
| w/o S1 S2 | 75.4 | 62.6 | 84.2 | 90.3 | 43.4 | 68.3 | 77.8 | 56.4 | 79.3 | 86.3 | 37.6 | 60.3 | 69.3 |
| CLIP-MoE | 74.6 | 65.0 | 86.0 | 92.0 | 46.8 | 71.7 | 80.4 | 60.5 | 82.3 | 88.8 | 42.1 | 64.7 | 73.2 |

Table 4: Ablation study on the impact of MCL stages 1 and 2 in CLIP-MoE performance.

**Computation and Data Efficiency** We compare the performance gains of our CLIP-MoE, trained on a 1M randomly sampled subset of Recap-DataComp-1B, to the CLIP-ViT-L-16-HTxt-Recap (Li et al., 2024b), which was trained from scratch on the entire Recap-DataComp-1B dataset. The activated parameter size of our CLIP-MoE, with 4 experts and top-2 routing, is 0.69B, which is comparable to the 0.64B parameter size of CLIP-ViT-L-16-HTxt-Recap. Thanks to MoE-Packing and leveraging the OpenAI CLIP dense checkpoint, our total training computation cost is less than 2% of that for CLIP-ViT-L-16-HTxt-Recap. As shown in Table 5, CLIP-MoE demonstrates comparable performance gains on retrieval tasks relative to CLIP-Recap, with even superior text-to-image retrieval performance on the Flickr30k dataset, highlighting the efficiency of our proposed MoE-Packing for CLIP. It is worth noting that CLIP-Recap uses an even larger text encoder.

| Model | COCO I2T @1 | @5 | @10 | COCO T2I @1 | @5 | @10 | Flickr I2T @1 | @5 | @10 | Flickr T2I @1 | @5 | @10 |
|-------|-----|-----|-----|-----|-----|-----|-----|-----|-----|-----|-----|-----|
| CLIP-MoE | +7.9 | +5.6 | +4.0 | +9.8 | +10.1 | +9.2 | +8.3 | +7.5 | +6.2 | +12.8 | +14.5 | +13.8 |
| CLIP-Recap | +10.8 | +7.7 | +5.5 | +12.3 | +12.3 | +10.7 | +10.9 | +8.3 | +6.8 | +11.9 | +12.9 | +11.9 |

Table 5: The performance gain of CLIP-MoE and CLIP-ViT-L-16-HTxt-Recap compared to the OpenAI CLIP-ViT-L-14 on the retrieval tasks

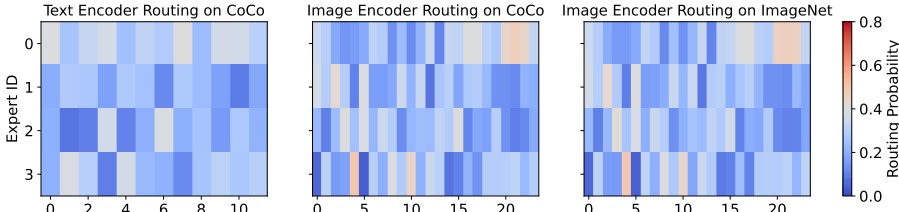

Figure 2: Proportion of tokens assigned to each expert on the COCO and ImageNet validation dataset. Here, we consider experts that are either selected as a first or second choice by the router.

**Routing analysis** To evaluate whether all the experts learned through MCL are utilized by CLIP-MoE, we perform an analysis of the routing strategy. We use the CLIP-MoE model with 4 experts

and top-2 routing trained on ShareGPT4V, and compute the proportion of tokens assigned to each expert. For retrieval tasks, we use the COCO validation dataset, and for zero-shot image classification, we use the ImageNet validation dataset. The analysis results are presented in Table 2. From the results, we observe that for experts from each MCL stage (represented by each column in the heatmap), there are consistently yellow areas (indicating heavily utilized experts). No column is entirely dark blue, which indicates that all MCL stages contribute useful experts to CLIP-MoE. This further validates the effectiveness of our MCL initialization in MoE-Packing.

**Case Study**

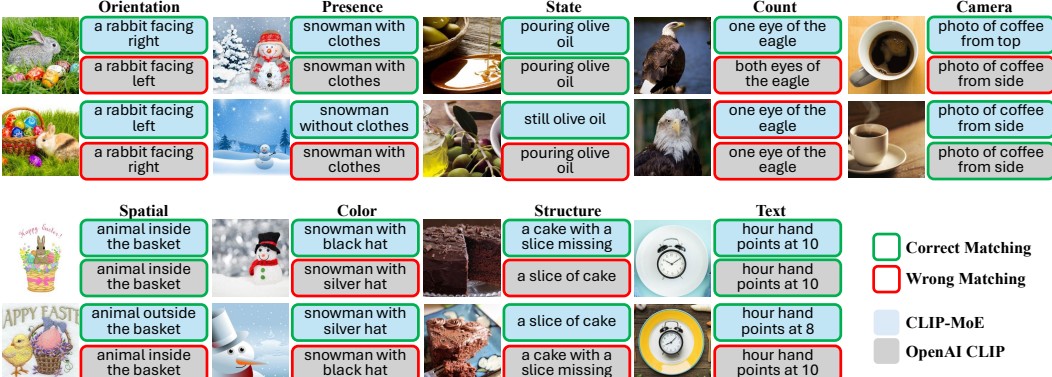

Figure 3: Example cases comparing the performance of CLIP-MoE and OpenAI CLIP on the MMVP-VLM Benchmark, illustrating differences in their ability to capture fine-grained semantic information.

We demonstrate the comparison between CLIP-MoE and OpenAI CLIP on samples from the MMVP-VLM Benchmark (Tong et al., 2024b). MMVP-VLM contains manually filtered image pairs with different semantics that are difficult to distinguish using the vanilla OpenAI CLIP. We task the models with matching the corresponding statement to the image. As shown in Figure 3, OpenAI CLIP struggles to distinguish fine-grained details in these image pairs. In cases like the alarm clock, OpenAI CLIP matches both images to the statement "hour hand points at 10." In other cases, such as the rabbit pair, OpenAI CLIP completely misinterprets the information and matches the opposite statement to the images. However, CLIP-MoE captures more fine-grained details and makes the correct match in most cases. It can accurately capture camera perspectives, as seen in the coffee example, orientation information in the rabbit example, and it demonstrates a superior ability to distinguish relations between objects, such as differentiating between "animal inside the basket" and "animal outside the basket."

## 6 CONCLUSION & FUTURE WORK

In this paper, we proposed a novel Diversified Multiplet Upcycling for CLIP to enhance the model with minimal computational overhead. Our method enables the extraction of diversified and complementary experts across multiple fine-tuning stages, which are then utilized within the MoE framework to capture richer information from the inputs. This approach is straightforward to apply, model-agnostic, and provides a new path to scale and improve CLIP foundation models. By leveraging off-the-shelf CLIP checkpoints and newly constructed high-quality image-text datasets, our method avoids the costly process of training CLIP models from scratch. We demonstrated the effectiveness and efficiency of our approach through extensive experiments across various datasets and tasks.

For future work, our current experiments are limited to image and text modalities. We plan to extend our method to additional modalities, such as audio and video. Beyond the fine-tuning settings explored in this paper, we aim to experiment with larger datasets and test large-scale continuous training settings to further explore the scalability and performance boundaries of Diversified Multiplet Upcycling. Additionally, while we tested CLIP-MoE as a vision encoder for MLLMs, we will also investigate its potential as a text encoder in generative tasks, such as in stable diffusion.

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
