# OpenReview forum: "CLIP-MoE: Towards Building Mixture of Experts for CLIP with Diversified Multiplet Upcycling"
_ICLR.cc/2025/Conference — Submitted to ICLR 2025_

### Official Review · Reviewer_nwTx · 2024-10-26

**Soundness:** 3
**Presentation:** 3
**Contribution:** 3
**Rating:** 5
**Confidence:** 4

**Summary:**

This paper proposed a novel Diversified Multiplet Upcycling for CLIP. Using relatively low computational cost, it trained a CLIP model with a Mixture of Experts (MoE) structure through Multistage Contrastive Learning, enabling the model to capture more fine-grained image information. The designed CLIP-MoE performs well in zero-shot retrieval, zero-shot image classification, and MLLM downstream tasks.

**Strengths:**

1. Achieved a Mixture of Experts (MoE) architecture for CLIP and proposed the Diversified Multiplet Upcycling method for CLIP.
2. Attained strong results with relatively low computational cost.
3. The paper provides extensive experimental results, evaluating CLIP-MoE’s performance in zero-shot retrieval, zero-shot image classification, and MLLM downstream tasks.

**Weaknesses:**

See Questions

**Questions:**

1. The paper claims that compared to the standard CLIP which often encodes inputs in a very coarse-grained manner, CLIP-MoE is able to encode richer information. I am curious about CLIP-MoE's performance in fine-grained image classification tasks, such as on the Stanford Cars, FGVC-Aircraft, Flowers102 and so on.

2. From the experimental results on LLaVA-1.5, replacing the CLIP in the LLaVA model with CLIP-MoE does not significantly improve performance across the five benchmarks; in fact, there are even some cases of performance decline. Additionally, the test results of the MLLM on benchmarks like MME and POPE fluctuate considerably. Are there other benchmark results, or perhaps reasoning cases for MLLM, that can further demonstrate CLIP-MoE’s ability to encode richer information in MLLM?

3. Why did you choose 4 experts? Is there a particular reason for this choice, and is there any observed relationship between the number of experts and CLIP-MoE’s performance?

I may reconsider my score based on your response to these issues.

---

> ### Author Response · Authors · 2024-11-21
> **Response to Reviewer nwTx**
>
> **W1:** The paper claims that compared to the standard CLIP which often encodes inputs in a very coarse-grained manner, CLIP-MoE is able to encode richer information. I am curious about CLIP-MoE’s performance in fine-grained image classification tasks, such as on the Stanford Cars, FGVC-Aircraft, Flowers102 and so on.
>
> **W2:** From the experimental results on LLaVA-1.5, replacing the CLIP in the LLaVA model with CLIP-MoE does not significantly improve performance across the five benchmarks; in fact, there are even some cases of performance decline. Additionally, the test results of the MLLM on benchmarks like MME and POPE fluctuate considerably. Are there other benchmark results, or perhaps reasoning cases for MLLM, that can further demonstrate CLIP-MoE’s ability to encode richer information in MLLM?
>
> **A1&A2:** Due to server migration, we will try to provide more evaluation on fine-grained image classification and MLLM before the discussion period ends.
>
> **W3:** Why did you choose 4 experts? Is there a particular reason for this choice, and is there any observed relationship between the number of experts and CLIP-MoE’s performance?
>
> **A3:** We chose 4 experts as it is the upper limit that allows training on 8×A100 GPUs while maintaining a reasonable batch size (8×100). Future work will explore configurations with more experts and evaluate their relationship to performance.

---

> > ### Comment · Reviewer_nwTx · 2024-11-25
> >
> > Thank you for your response. Since you have not provided additional experimental results, my previous concerns remain unresolved. The existing experimental results are insufficient to convince me of the innovation and effectiveness of CLIP-MoE. If the authors are still unable to address my concerns, I will adjust my score accordingly.

---

> > > ### Author Response · Authors · 2024-12-01
> > > **Reply to Reviewer nwTx**
> > >
> > > For fine-grained image classification tasks, we provide additional evaluations on the Stanford Cars and FGVC-Aircraft datasets. While CLIP-MoE demonstrates better performance on FGVC-Aircraft, it performs slightly worse on Stanford Cars. It is important to note that capturing more diverse information does not necessarily translate into improved zero-shot image classification accuracy on every benchmark. Our primary contribution lies in enabling the model to encode richer and more diverse information, rather than solely optimizing for zero-shot classification performance.
> > >
> > > Additionally, CLIP-MoE significantly outperforms other approaches such as Direct Fine-Tuning, Upcycling, and LongCLIP, as shown in the results below:
> > >
> > > | **Dataset**        | **OpenAI** | **CLIP-MoE** | **Direct-FT** | **Upcycling** | **Longclip** |
> > > |---------------------|------------|--------------|---------------|---------------|--------------|
> > > | **FGVC-Aircraft**   | 0.478      | **0.498**    | 0.297         | 0.332         | 0.470        |
> > > | **Stanford Cars**   | **0.763**  | 0.746        | 0.513         | 0.609         | 0.730        |
> > >
> > > For further MLLM evaluation results, we regret to inform you that due to hardware constraints, we are unable to add additional evaluations at this time. These evaluations will be included in a future version of the paper. However, we provide supplementary results on the MMVP[1] benchmark, which further illustrate that the experts learned by CLIP-MoE effectively capture diverse and complementary information. The MMVP benchmark requires the CLIP model to select the correct image based on a textual statement from a pair of visually similar images. The evaluation data are carefully filtered into nine distinct attributes by human annotators.
> > >
> > > The results below clearly show that different experts specialize in different attributes. For example, Expert0 performs best on attributes such as Presence of Specific Features, Quantity and Count, Color and Appearance, and Viewpoint and Perspective. Expert1 excels in Structural and Physical Characteristics. Expert2 focuses on Orientation and Direction, State and Condition, and Texts, while Expert3 specializes in Orientation and Direction, as well as Viewpoint and Perspective.
> > >
> > > These results highlight the effectiveness of MCL, as it successfully generates experts that specialize in capturing diverse and complementary information.
> > >
> > > | Attribute                       | Expert0 | Expert1 | Expert2 | Expert3 |
> > > |---------------------------------|---------|---------|---------|---------|
> > > | Orientation and Direction       | 40      | 33.3    | **46.7**    | **46.7**    |
> > > | Presence of Specific Features   | **33.3**    | 26.7    | 26.7    | 13.3    |
> > > | State and Condition             | 20      | 40      | **53.3**    | 40      |
> > > | Quantity and Count              | **60**      | 46.7    | 40      | 40      |
> > > | Positional and Relational Context | **46.7**  | 33.3    | 40      | 26.7    |
> > > | Color and Appearance            | **26.7**    | 13.3    | 6.7     | 6.7     |
> > > | Structural and Physical Characteristics | 26.7 | **46.7**    | 40      | 33.3    |
> > > | Texts                           | 26.7    | 40      | **46.7**    | 40      |
> > > | Viewpoint and Perspective       | 53.3      | 46.7    | 40      | **60**    |
> > >
> > > [1]Tong, S., Liu, Z., Zhai, Y., Ma, Y., LeCun, Y., & Xie, S. (2024). Eyes wide shut? exploring the visual shortcomings of multimodal llms. In Proceedings of the IEEE/CVF Conference on Computer Vision and Pattern Recognition (pp. 9568-9578).

---

> > > ### Author Response · Authors · 2024-12-02
> > >
> > > Thank you for your thoughtful feedback and suggestions throughout the review process. I wanted to kindly draw your attention to the additional experiments we’ve included in the latest revision, which address the points raised and further strengthen our findings. Your insights on these additions would be highly valuable to us.

---

### Official Review · Reviewer_fub1 · 2024-11-02

**Soundness:** 2
**Presentation:** 3
**Contribution:** 2
**Rating:** 5
**Confidence:** 4

**Summary:**

The paper presents a novel model-agnostic fine-tuning strategy called Diversified Multiplet Upcycling (DMU) for enhancing the CLIP framework. This approach utilizes the sparsely activated Mixture of Experts (MoE) architecture to extend model capacity without the need for training from scratch, instead leveraging pre-trained dense checkpoints. DMU fine-tunes a base dense CLIP model to create multiple multiplet experts through Multistage Contrastive Learning (MCL), which encodes diverse information via clustering. These multiplet models share all parameters except for the feed-forward network (FFN) layers, resulting in specialized FFN experts that capture different input aspects.

**Strengths:**

1. It introduces a new perspective on fine-tuning CLIP with Multistage Contrastive Learning (MCL) and validates this method through proper experiments.

2. The results show improvements across various downstream tasks, indicating its potential effectiveness in practical applications, supported by coherent writing and a relative reasonable experimental setup.

**Weaknesses:**

1. A substantive assessment of the weaknesses of the paper. Focus on constructive and actionable insights on how the work could improve towards its stated goals. Be specific, avoid generic remarks. For example, if you believe the contribution lacks novelty, provide references and an explanation as evidence; if you believe experiments are insufficient, explain why and exactly what is missing, etc.

2. Firstly, there is a mislabeling in the paper. In Section 5.5, Table 1 presents the results of the text-to-image retrieval experiment on the COCO dataset. The CLIP-MoE model trained on the Recap-DC dataset achieved an @10 value of 79.4, which is lower than the Upcycling's 79.9. However, the value of 79.4 is erroneously bolded, which may lead to confusion. It is recommended that this be corrected to avoid misinterpretation of the results.

3. Secondly, it might be beneficial to include an analysis of the number of experts utilized in the method. Currently, the CLIP-MoE employs 4 experts to balance performance and computational cost. However, conducting experiments to demonstrate how varying the number of experts—either increasing or decreasing—affects both performance and computational efficiency could provide more compelling evidence for the chosen configuration.

4. Thirdly, while Multistage Contrastive Learning is described in the paper as enabling experts to learn different aspects of features—such as color, texture, and shape—this explanation seems to lack supporting experimental results. To strengthen this claim, it would be beneficial to provide concrete evidence, such as demonstrating that models trained at different stages exhibit distinct capabilities on datasets characterized by these specific features. Alternatively, offering a more nuanced explanation of the effectiveness of Multistage Contrastive Learning would help clarify its role and prevent potential misunderstandings.

5. Lastly, I think the explanation in the ablation study section to be unclear. This section mentions training a CLIP-MoE model with two experts, while Table 4 compares a fully trained CLIP-MoE with a model that has not undergone the first and second stages of MCL training. First, is the latter the CLIP-MoE model with two experts mentioned in the paper? Additionally, based on my understanding, since the fully trained CLIP-MoE in the paper includes 4 experts, it should undergo 4 stages of MCL training. The results related to the CLIP-MoE in Table 4 are also obtained after 4 stages of MCL training. Why is a model that has not gone through the first and second stages of training used for comparison?

**Questions:**

Please see the weakness section.

---

> ### Author Response · Authors · 2024-11-21
> **Response to Reviewer fub1**
>
> **W1:** Firstly, there is a mislabeling in the paper. In Section 5.5, Table 1 presents the results of the text-to-image retrieval experiment on the COCO dataset. The CLIP-MoE model trained on the Recap-DC dataset achieved an @10 value of 79.4, which is lower than the Upcycling's 79.9. However, the value of 79.4 is erroneously bolded, which may lead to confusion. It is recommended that this be corrected to avoid misinterpretation of the results.
>
> **A1:** We sincerely apologize for the incorrect bolding in the table. Thank you for bringing this to our attention. We will ensure this error is corrected in the final version to avoid any confusion.
>
> **W2:** Secondly, it might be beneficial to include an analysis of the number of experts utilized in the method. Currently, the CLIP-MoE employs 4 experts to balance performance and computational cost. However, conducting experiments to demonstrate how varying the number of experts—either increasing or decreasing—affects both performance and computational efficiency could provide more compelling evidence for the chosen configuration.
>
> **A2:** Thank you for your insightful suggestion. We selected 4 experts as it represents the upper limit that allows training on 8×A100 GPUs while maintaining a reasonable batch size (8×100). Exploring configurations with more or fewer experts and analyzing their relationship to performance and computational efficiency is a promising direction for future work.
>
> **W3:** Thirdly, while Multistage Contrastive Learning is described in the paper as enabling experts to learn different aspects of features—such as color, texture, and shape—this explanation seems to lack supporting experimental results. To strengthen this claim, it would be beneficial to provide concrete evidence, such as demonstrating that models trained at different stages exhibit distinct capabilities on datasets characterized by these specific features. Alternatively, offering a more nuanced explanation of the effectiveness of Multistage Contrastive Learning would help clarify its role and prevent potential misunderstandings.
>
> **A3:** MCL aims to generate experts with distinct expertise through iterative clustering and training. At a high level, the clustering results of each stage reflect the feature distributions learned so far, while the MCL loss encourages the model to learn new distributions distinct from the existing ones. We do not impose assumptions on the clustering outcomes, and as noted in Figure 1, terms like "color," "shape," and "texture" are metaphorical representations of abstract features. In practice, clustering results may not always align with human-intuitive categories. However, as shown in the original MCL paper[1], experiments on synthetic datasets demonstrate that clustering can yield interpretable groupings, validating MCL's effectiveness.
> To further support that MCL can produce experts with different focus, we will provide the evaluation of the obtained experts on the MMVP[2] benchmark later, which contains ten manually defined attributes.
>
> **W4:** Lastly, I think the explanation in the ablation study section to be unclear. This section mentions training a CLIP-MoE model with two experts, while Table 4 compares a fully trained CLIP-MoE with a model that has not undergone the first and second stages of MCL training. First, is the latter the CLIP-MoE model with two experts mentioned in the paper? Additionally, based on my understanding, since the fully trained CLIP-MoE in the paper includes 4 experts, it should undergo 4 stages of MCL training. The results related to the CLIP-MoE in Table 4 are also obtained after 4 stages of MCL training. Why is a model that has not gone through the first and second stages of training used for comparison?
>
> **A4:** The ablation study aims to evaluate how much of the performance gain stems from our proposed MCL initialization. CLIP-MoE (w/o S1, S2) integrates the original dense CLIP checkpoint with FFN fine-tuned on ShareGPT4V (MCL-Stage0). In contrast, CLIP-MoE incorporates FFNs from the original dense CLIP checkpoint, MCL-Stage0, MCL-Stage1, and MCL-Stage2. This demonstrates the incremental benefits of MCL stages, which further validates the MCL initialization in DMU. We acknowledge that the explanation in the current version of the paper could be clearer, and we will revise this section in the final version to improve clarity and address these concerns.
>
> [1]Zhang, J., Lan, X., Qu, X., Cheng, Y., Feng, M., \& Hooi, B. (2025). Learning the Unlearned: Mitigating Feature Suppression in Contrastive Learning. In European Conference on Computer Vision (pp. 35-52). Springer, Cham.
>
> [2]Tong, S., Liu, Z., Zhai, Y., Ma, Y., LeCun, Y., \& Xie, S. (2024). Eyes wide shut? exploring the visual shortcomings of multimodal llms. In Proceedings of the IEEE/CVF Conference on Computer Vision and Pattern Recognition (pp. 9568-9578).

---

> > ### Author Response · Authors · 2024-12-03
> >
> > Hi, for **W3** please kindly see our additional evaluation on the MMVP benchmark, which clearly shows that **models trained at different stages exhibit distinct capabilities**.
> >
> > The MMVP benchmark requires the CLIP model to select the correct image based on a textual statement from a pair of visually similar images. The evaluation data are carefully filtered into nine distinct attributes by human annotators.
> >
> > The results below clearly show that different experts specialize in different attributes. For example, Expert0 performs best on attributes such as Presence of Specific Features, Quantity and Count, Color and Appearance, and Viewpoint and Perspective. Expert1 excels in Structural and Physical Characteristics. Expert2 focuses on Orientation and Direction, State and Condition, and Texts, while Expert3 specializes in Orientation and Direction, as well as Viewpoint and Perspective.
> >
> > These results highlight the effectiveness of MCL, as it successfully generates experts that specialize in capturing diverse and complementary information.
> >
> > | Attribute                       | Expert0 | Expert1 | Expert2 | Expert3 |
> > |---------------------------------|---------|---------|---------|---------|
> > | Orientation and Direction       | 40      | 33.3    | **46.7**    | **46.7**    |
> > | Presence of Specific Features   | **33.3**    | 26.7    | 26.7    | 13.3    |
> > | State and Condition             | 20      | 40      | **53.3**    | 40      |
> > | Quantity and Count              | **60**      | 46.7    | 40      | 40      |
> > | Positional and Relational Context | **46.7**  | 33.3    | 40      | 26.7    |
> > | Color and Appearance            | **26.7**    | 13.3    | 6.7     | 6.7     |
> > | Structural and Physical Characteristics | 26.7 | **46.7**    | 40      | 33.3    |
> > | Texts                           | 26.7    | 40      | **46.7**    | 40      |
> > | Viewpoint and Perspective       | 53.3      | 46.7    | 40      | **60**    |

---

> ### Author Response · Authors · 2024-12-02
>
> Thank you for your thoughtful feedback and suggestions throughout the review process. I wanted to kindly draw your attention to the additional experiments we’ve included in the latest revision, which address the points raised and further strengthen our findings. Your insights on these additions would be highly valuable to us.

---

### Official Review · Reviewer_a1CZ · 2024-11-03

**Soundness:** 3
**Presentation:** 2
**Contribution:** 2
**Rating:** 6
**Confidence:** 3

**Summary:**

This work applies the recent popular concept of Mixture of Experts (MoE) in LLMs to CLIP. Specifically, it first uses Multistage Contrastive Learning (MCL) to lightly train various diversified FFNs. Then, these FFNs are used as experts in a CLIP-MoE, and the router is fine-tuned to obtain the final model. The work achieves impressive results in zero-shot image-text retrieval and zero-shot image classification tasks.

**Strengths:**

1. This work claims to be the first attempt to apply MoE to a CLIP-style model. I also believe it is an early attempt at incorporating MoE into contrastive learning.
2. The authors propose a novel method for initializing multiple experts in MoE, using MCL to first train and obtain diverse yet meaningful experts.
3. The proposed method is efficient, as it does not require retraining from scratch.
4. The paper is well-written, allowing me to quickly understand the authors’ key ideas.

**Weaknesses:**

1. The motivation behind this work—that CLIP often encodes inputs in a very coarse-grained manner—is neither novel nor particularly compelling. Additionally, the paper lacks an in-depth analysis of why this issue arises and does not clearly explain how the proposed method addresses it. The introduction could benefit from further refinement, as it currently seems to have chosen a weak motivation simply to justify applying MoE to CLIP.

2. The rationale for using Multistage Contrastive Learning (MCL) is not fully convincing. The paper suggests that clustering can simulate grouping raw data by attributes, such as color or shape. However, as clustering is conducted entirely within CLIP’s feature space, this outcome is not necessarily assured. The authors could strengthen their argument by visualizing each expert’s focus on distinct details, using methods such as t-SNE. Additionally, the routing analysis and case study presented in the experiments are too general to effectively illustrate MCL’s advantages. Beside, a more targeted comparison that highlights each expert's independent downstream task performance would provide clearer insights into the effectiveness of MCL.

3. The zero-shot tasks for CLIP lack sufficient comparisons to other state-of-the-art methods. This work mainly compares against Long-CLIP, which was primarily designed for sequence extension rather than enhancing CLIP’s zero-shot performance in downstream tasks.

4. The design of the ablation study is somewhat unclear. The sole comparison (w/o S1, S2) introduces too many variables, such as the number of experts, whether the experts were fine-tuned, and whether fine-tuning occurred on the original dataset or the clustered data (assuming clustering aims to mine hard negatives). The authors should design more distinct ablation studies to clearly demonstrate the contribution of each component.

5. Minor typo: in line 488, "Table 2" should refer to "Figure 2."

**Questions:**

1. In the MCL process, is each FFN initialized from the previous stage’s FFN or from the pre-trained CLIP FFN?

2.  Why does upcycling appear less effective than direct fine-tuning in Table 1, yet w/o S1 S2 in Table 4, it outperforms both methods significantly, such as on Flickr I2T?

---

> ### Author Response · Authors · 2024-11-21
> **Response to Reviewer a1CZ (1)**
>
> **W1:** The motivation behind this work—that CLIP often encodes inputs in a very coarse-grained manner—is neither novel nor particularly compelling. Additionally, the paper lacks an in-depth analysis of why this issue arises and does not clearly explain how the proposed method addresses it. The introduction could benefit from further refinement, as it currently seems to have chosen a weak motivation simply to justify applying MoE to CLIP.
>
> **A1:** Thank you for your valuable comments. We believe that the coarse-grained encoding in CLIP can be attributed to feature suppression in contrastive learning. Feature suppression occurs when, due to competing features, the model prioritizes certain features while ignoring others. Although the reasons for this phenomenon are complex and covered in prior works[1,2,3], our primary focus is not on feature. Thus, we provided limited detail on this aspect in the introduction.  Our proposed CLIP-MoE is designed not only to mitigate feature suppression but also to introduce several additional advantages. Through our Diversified Multiplet Upcycling (DMU) approach, CLIP-MoE flexibly updates pre-trained CLIP models with high-quality data, maintains the original model's performance, improves performance on tasks related to new data, and avoids the need for training from scratch. We acknowledge that the introduction could be further refined to include more background and clearly articulate our motivation.
>
> **W2:** The rationale for using Multistage Contrastive Learning (MCL) is not fully convincing. The paper suggests that clustering can simulate grouping raw data by attributes, such as color or shape. However, as clustering is conducted entirely within CLIP’s feature space, this outcome is not necessarily assured. The authors could strengthen their argument by visualizing each expert’s focus on distinct details, using methods such as t-SNE. Additionally, the routing analysis and case study presented in the experiments are too general to effectively illustrate MCL’s advantages. Besides, a more targeted comparison that highlights each expert's independent downstream task performance would provide clearer insights into the effectiveness of MCL.
>
> **A2:** MCL aims to generate experts with distinct expertise through iterative clustering and training. At a high level, the clustering results of each stage reflect the feature distributions learned so far, while the MCL loss encourages the model to learn new distributions distinct from the existing ones. We do not impose assumptions on the clustering outcomes, and as noted in Figure 1, terms like "color," "shape," and "texture" are metaphorical representations of abstract features. In practice, clustering results may not always align with human-intuitive categories. However, as shown in the original MCL paper[1], experiments on synthetic datasets demonstrate that clustering can yield interpretable groupings, validating MCL's effectiveness. To further support that MCL can produce experts with different focus, we will provide the evaluation of the obtained experts on the MMVP benchmark[4] later, which contains ten manually defined attributes.
>
> **W3:** The zero-shot tasks for CLIP lack sufficient comparisons to other state-of-the-art methods. This work mainly compares against Long-CLIP, which was primarily designed for sequence extension rather than enhancing CLIP’s zero-shot performance in downstream tasks.
>
> **A3:** Our primary claim is that DMU can convert a dense CLIP checkpoint into a sparse MoE architecture with performance improvements. Thus, our comparisons mainly focus on the base dense checkpoint. To further validate our method, we plan to extend our experiments to additional dense checkpoints in future work.
> We chose Long-CLIP as a baseline for two reasons: (1) it also enhances a pre-trained CLIP checkpoint without training from scratch, similar to our approach; and (2) it allows for fair comparisons by using the same training data. Many state-of-the-art CLIP models rely on superior data, making direct comparison challenging. However, these models can serve as new base checkpoints for CLIP-MoE in future work. Additionally, we include CLIP-ViT-L-16-HTxt-Recap as a performance reference in Section 5.6.

---

> ### Author Response · Authors · 2024-11-21
> **Response to Reviewer a1CZ (2)**
>
> **W4:** The design of the ablation study is somewhat unclear. The sole comparison (w/o S1, S2) introduces too many variables, such as the number of experts, whether the experts were fine-tuned, and whether fine-tuning occurred on the original dataset or the clustered data (assuming clustering aims to mine hard negatives). The authors should design more distinct ablation studies to clearly demonstrate the contribution of each component.
>
> **A4:** The ablation study aims to evaluate how much of the performance gain stems from our proposed MCL initialization. CLIP-MoE (w/o S1, S2) integrates the original dense CLIP checkpoint with FFN fine-tuned on ShareGPT4V (MCL-Stage0). In contrast, CLIP-MoE incorporates FFNs from the original dense CLIP checkpoint, MCL-Stage0, MCL-Stage1, and MCL-Stage2. This demonstrates the incremental benefits of MCL stages, which further validates the MCL initialization in DMU.
>
> **Q1:** In the MCL process, is each FFN initialized from the previous stage’s FFN or from the pre-trained CLIP FFN?
>
> **AQ1:** Each FFN is initialized from the pre-trained CLIP FFN (dense checkpoint).  This follows the methodology outlined in MCL[1]. Initializing from the pre-trained checkpoint allows each stage to learn new and distinct features without being constrained by previously learned properties in previous stages. Retaining prior properties could hinder the model's ability to acquire new information.
>
> **Q2:** Why does upcycling appear less effective than direct fine-tuning in Table 1, yet w/o S1 S2 in Table 4, it outperforms both methods significantly, such as on Flickr I2T?
>
> **AQ2:** As mentioned above, CLIP-MoE (w/o S1, S2) integrates the original CLIP dense checkpoint with FFNs fine-tuned on ShareGPT4V (MCL-Stage0). This differs from both upcycling and direct fine-tuning. Upcycling is primarily designed for larger-scale continuous training settings. In our experiments, the dataset may not have been sufficient to fully support upcycling’s ability to learn enough diversified experts, which could explain why it does not significantly outperform direct fine-tuning in this context.
>
> [1]Zhang, J., Lan, X., Qu, X., Cheng, Y., Feng, M., \& Hooi, B. (2025). Learning the Unlearned: Mitigating Feature Suppression in Contrastive Learning. In European Conference on Computer Vision (pp. 35-52). Springer, Cham.
>
> [2]Bleeker, M., Yates, A., \& de Rijke, M. (2022). Reducing Predictive Feature Suppression in Resource-Constrained Contrastive Image-Caption Retrieval. arXiv preprint arXiv:2204.13382.
>
> [3]Robinson, J., Sun, L., Yu, K., Batmanghelich, K., Jegelka, S., \& Sra, S. (2021). Can contrastive learning avoid shortcut solutions?. Advances in neural information processing systems, 34, 4974-4986.
>
> [4]Tong, S., Liu, Z., Zhai, Y., Ma, Y., LeCun, Y., \& Xie, S. (2024). Eyes wide shut? exploring the visual shortcomings of multimodal llms. In Proceedings of the IEEE/CVF Conference on Computer Vision and Pattern Recognition (pp. 9568-9578).

---

> > ### Comment · Reviewer_a1CZ · 2024-11-22
> >
> > Thank you for your detailed responses. However, my core concerns remain inadequately addressed:
> >
> > * I hope to see a more detailed analysis or experimental evidence supporting the motivation for applying MoE to CLIP. Without such clarification, the motivation feels somewhat weak and unconvincing.
> >
> > * There is still a lack of in-depth experiments to understand the roles of different experts. Reporting a general performance metric alone does not illustrate the benefits of using MoE. For example, what advantages do these complex designs offer compared to simpler augmentation techniques, such as ensemble models built from multiple random seeds?
> >
> > * There is insufficient comparison with strong baselines. While I understand the paper focuses on comparing with dense CLIP checkpoints, the additional training steps warrant comparison with stronger methods that also aim to improve the original CLIP model.
> >
> > In your rebuttal, these issues were not thoroughly addressed, and most of the claims indicate that refinements or additional experiments will be provided in future work. As such, I will keep my original score unchanged.

---

> > > ### Author Response · Authors · 2024-12-01
> > > **Reply to Reviewer a1CZ (3)**
> > >
> > > **"There is still a lack of in-depth experiments to understand the roles of different experts."**
> > >
> > > Sorry for the lack of clarification. To address this, we provide additional evaluation results on the MMVP benchmark, which demonstrate that different experts capture diverse and complementary information. The MMVP benchmark requires the CLIP model to select the correct image based on a textual statement from a pair of visually similar images. The evaluation data are carefully filtered into nine distinct attributes by human annotators.
> > >
> > > The results below clearly show that different experts specialize in different attributes. For example, Expert0 performs best on attributes such as Presence of Specific Features, Quantity and Count, Color and Appearance, and Viewpoint and Perspective. Expert1 excels in Structural and Physical Characteristics. Expert2 focuses on Orientation and Direction, State and Condition, and Texts, while Expert3 specializes in Orientation and Direction, as well as Viewpoint and Perspective.
> > >
> > > These results highlight the effectiveness of MCL, as it successfully generates experts that specialize in capturing diverse and complementary information.
> > >
> > > | Attribute                       | Expert0 | Expert1 | Expert2 | Expert3 |
> > > |---------------------------------|---------|---------|---------|---------|
> > > | Orientation and Direction       | 40      | 33.3    | **46.7**    | **46.7**    |
> > > | Presence of Specific Features   | **33.3**    | 26.7    | 26.7    | 13.3    |
> > > | State and Condition             | 20      | 40      | **53.3**    | 40      |
> > > | Quantity and Count              | **60**      | 46.7    | 40      | 40      |
> > > | Positional and Relational Context | **46.7**  | 33.3    | 40      | 26.7    |
> > > | Color and Appearance            | **26.7**    | 13.3    | 6.7     | 6.7     |
> > > | Structural and Physical Characteristics | 26.7 | **46.7**    | 40      | 33.3    |
> > > | Texts                           | 26.7    | 40      | **46.7**    | 40      |
> > > | Viewpoint and Perspective       | 53.3      | 46.7    | 40      | **60**    |
> > >
> > >
> > > **"There is insufficient comparison with strong baselines."**
> > >
> > > It’s important to note that our additional training cost is less than 2% of the total training cost of the original CLIP. If there are similar upcycling methods you are aware of that would provide a fair baseline for comparison, we would be happy to include them in a future version of the paper.

---

> > > > ### Comment · Reviewer_a1CZ · 2024-12-01
> > > >
> > > > Thank the authors for the newly added experiments exploring the roles of different experts.
> > > >
> > > > I greatly appreciate this experiment, which strongly demonstrates the effectiveness of MOE. I suggest leveraging this experiment to further strengthen the motivation of the paper.
> > > >
> > > > However, since the revision deadline has passed, I am unable to see the refined introduction. Therefore, I will raise my score by one point accordingly, but I cannot increase it further, as the current introduction still feels like it requires significant revisions.

---

### Official Review · Reviewer_ihbw · 2024-11-04

**Soundness:** 3
**Presentation:** 3
**Contribution:** 2
**Rating:** 5
**Confidence:** 4

**Summary:**

This work introduces a novel method, Diversified Multiplet Upcycling (DMU), for constructing Mixture of Experts (MoE) on CLIP models. The study showcases the effectiveness of DMU by presenting results on zero-shot image retrieval benchmarks, zero-shot image classification benchmarks, and MLLM understanding benchmarks by integrating it as the vision encoder in Llava-1.5. The findings demonstrate that DMU is indeed effective in enhancing the performance of CLIP models.

**Strengths:**

The experiments conducted in this study cover a wide range of tasks including zero-shot retrieval, classification, and MLLM understanding. By comparing against the original OpenAI CLIP large model, LongCLIP, and Llava-1.5, the results demonstrate superior performance of the proposed method. Additionally, this work shows that the Diversified Multiplet Upcycling (DMU) is more efficient compared to simply upcycling the FFN to a MoE.

**Weaknesses:**

1. The zero-shot image classification results should be more diverse. Only including ImageNet, ImageNet-O, ImageNet-V2, CIFAR-10, and CIFAR-100 is not sufficient. Please refer to the CLIP benchmark [1]. I believe that including ImageNet, ImageNetV2, ImageNet-A, ImageNet-R, ImageNet-Sketch, and ObjectNet datasets is essential for a more comprehensive evaluation.

2. I'm interested in the performance on the MM-Vet benchmark.


Reference:
[1] https://github.com/LAION-AI/CLIP_benchmark

**Questions:**

please see weaknesses

---

> ### Author Response · Authors · 2024-11-21
> **Response to Reviewer ihbw**
>
> **W1:** The zero-shot image classification results should be more diverse. Only including ImageNet, ImageNet-O, ImageNet-V2, CIFAR-10, and CIFAR-100 is not sufficient. Please refer to the CLIP benchmark [1]. I believe that including ImageNet, ImageNetV2, ImageNet-A, ImageNet-R, ImageNet-Sketch, and ObjectNet datasets is essential for a more comprehensive evaluation.
>
> **A1:** Our comparison on zero-shot image classification follows the protocols in Recap-1B[1] and Long-CLIP[2]. Furthermore, as MLLMs are gaining prominence, recent works[3,4] suggest that testing within the MLLM framework may provide a more comprehensive evaluation. We appreciate your valuable feedback and will include evaluations from the CLIP benchmark in the final version of the paper. We will try to provide more evaluation on zero-shot image classification before the discussion period ends.
>
> **W2:** I’m interested in the performance on the MM-Vet benchmark.
>
> **A2:** Thank you for raising this point. Due to server migration, we will try to provide an evaluation of the MM-Vet benchmark before the discussion period ends. A full evaluation will be included in the final version of the paper.
>
> [1]Li, X., Tu, H., Hui, M., Wang, Z., Zhao, B., Xiao, J., ... \& Xie, C. (2024). What If We Recaption Billions of Web Images with LLaMA-3?. arXiv preprint arXiv:2406.08478.
>
> [2]Zhang, B., Zhang, P., Dong, X., Zang, Y., \& Wang, J. (2025). Long-clip: Unlocking the long-text capability of clip. In European Conference on Computer Vision (pp. 310-325). Springer, Cham.
>
> [3]Tong, S., Liu, Z., Zhai, Y., Ma, Y., LeCun, Y., \& Xie, S. (2024). Eyes wide shut? exploring the visual shortcomings of multimodal llms. In Proceedings of the IEEE/CVF Conference on Computer Vision and Pattern Recognition (pp. 9568-9578).
>
> [4]Tong, S., Brown, E., Wu, P., Woo, S., Middepogu, M., Akula, S. C., ... \& Xie, S. (2024). Cambrian-1: A fully open, vision-centric exploration of multimodal llms. arXiv preprint arXiv:2406.16860.

---

### Author Response · Authors · 2024-12-03
**General Response**

We sincerely thank all the reviewers for their time, thoughtful feedback, and constructive suggestions. We are encouraged that the reviewers recognize the following:

- **Novelty**: Our proposed Diversified Multiplet Upcycling (DMU) is novel (Reviewers ihbw, a1CZ, fub1, nwTx) and represents an early exploration of incorporating MoE into contrastive learning (Reviewer a1CZ).
- **Strong Experiments**: Our experimental results are strong and comprehensive (Reviewers ihbw, a1CZ, fub1, nwTx), demonstrating the efficiency of our proposed DMU (Reviewers ihbw, a1CZ, nwTx).
- **Clarity**: Our paper is well-written and easy to follow (Reviewer a1CZ).

We have addressed most of the reviewers' concerns during the discussion period. Below, we summarize our responses to common questions:

### **More Evaluation Results on Zero-Shot Image Classification**
We have included additional evaluations on ImageNet-A, ImageNet-R, ImageNet-Sketch, Stanford Cars, and FGVC-Aircraft. However, it is important to note that capturing more diverse information does not always result in improved zero-shot classification accuracy, particularly on benchmarks designed to test robustness. In some cases, encoding richer information may reduce robustness. Our primary contribution lies in enhancing the model’s ability to encode richer and more diverse information, rather than solely optimizing for zero-shot classification performance. Furthermore, CLIP-MoE significantly outperforms other approaches such as Direct Fine-Tuning, Upcycling, and LongCLIP.

### **Evidence for Diverse and Complementary Expert Contributions**
To demonstrate that different experts capture diverse and complementary information, we have provided additional evaluation results on the MMVP benchmark. These results clearly show that different experts specialize in distinct attributes, validating that DMU successfully enables the creation of diverse and complementary experts.

We deeply appreciate the reviewers' valuable feedback and suggestions, which have greatly improved our work.

---

### Meta-Review · Area_Chair_UsbN · 2024-12-20

**Metareview:**

(a) Scientific Claims and Findings

The paper introduces Diversified Multiplet Upcycling (DMU), a novel method for enhancing CLIP models using a Mixture of Experts (MoE) architecture. By employing Multistage Contrastive Learning (MCL), DMU aims to improve the model's capacity to capture fine-grained image information without retraining from scratch. The method is evaluated on zero-shot image retrieval, classification, and MLLM understanding benchmarks, demonstrating good performance compared to existing models. Reviewers highlight the method's potential to enhance CLIP's performance across various tasks.

(b) Strengths

Reviewer ihbw notes the comprehensive range of tasks covered in the experiments, demonstrating the method's effectiveness. a1CZ appreciates the novel application of MoE to CLIP and the efficient training strategy that avoids retraining from scratch. fub1 commends the new perspective on fine-tuning CLIP with MCL and the coherent writing. nwTx highlights the strong results achieved with relatively low computational cost and the extensive experimental evaluation.

(c) Weaknesses

The reviewers identify several weaknesses. ihbw suggests expanding the range of datasets for a more comprehensive evaluation. a1CZ questions the motivation and rationale behind the method, suggesting a need for deeper analysis and more targeted comparisons. fub1 points out mislabeling in the paper and suggests analyzing the number of experts used. nwTx raises concerns about the performance in fine-grained tasks and the choice of the number of experts, suggesting that more benchmarks or reasoning cases could further demonstrate the method's effectiveness.

(d) Decision Reasons

The decision to reject the paper is based on concerns about the depth of analysis and the clarity of the method's motivation, as highlighted by reviewers a1CZ and fub1. While the paper presents a compelling approach, the lack of comprehensive dataset evaluation and the need for more detailed comparisons and analyses reduce its impact. Additionally, the questions about the choice of the number of experts and the performance in fine-grained tasks, as noted by nwTx, further weaken the case for acceptance. Despite the strengths in experimental validation and efficiency, the weaknesses in motivation, analysis, and evaluation lead to the decision to reject.

**Additional Comments On Reviewer Discussion:**

During the rebuttal period, the authors addressed some concerns raised by the reviewers, but the responses did not lead to significant changes in their evaluations.

Reviewer ihbw appreciated the quick response but maintained concerns about the diversity of the evaluation and the motivation for applying MoE to CLIP. They decided to keep their original score, looking for supplemental results and further analysis to address these points.

Reviewer a1CZ acknowledged the newly added experiments exploring the roles of different experts, which they found valuable. However, they noted that the core concerns about the motivation and comparison with strong baselines were not thoroughly addressed. While they raised their score by one point due to the new experiments, they felt the introduction still required significant revisions and maintained that the paper did not meet the quality standards for ICLR.

Reviewer fub1 found that the paper still had issues to be fixed in the next version and decided to maintain their score, indicating that the rebuttal did not sufficiently address their concerns.

Reviewer nwTx noted that the additional experimental results provided in the rebuttal were insufficient to resolve their concerns about the innovation and effectiveness of CLIP-MoE. They decided to adjust their score accordingly, indicating a lack of conviction in the paper's contributions.

In the final discussion, after the rebuttal phase, reviewers a1CZ, nwTx, and ihbw agreed on recommending rejection. They cited the lack of clear motivation, insufficient innovation, and the inability of the additional experimental results to convincingly demonstrate the advantages of applying MoE to CLIP models. The consensus was that the paper did not meet the standards required for ICLR, leading to the decision to reject.

---

### Decision · Program_Chairs · 2025-01-22

Reject